# A Subtlety of Sizing the Inset Gap Width of a Microstrip Antenna When Built on an Ultra-Thin Substrate in the S-Band

**DOI:** 10.3390/s23010213

**Published:** 2022-12-25

**Authors:** Miroslav Joler, Leo Mihalić

**Affiliations:** Faculty of Engineering, University of Rijeka, 51000 Rijeka, Croatia

**Keywords:** ultra-thin substrate, rectangular microstrip antenna, inset feed, Pyralux, quarter-wave transformer

## Abstract

In this paper, Pyralux—a modern, ultra-thin, and acrylic-based laminate—was tested as a substrate of a microstrip antenna to examine the antenna characteristics when it is built on such a thin, flexible, and robust dielectric material, with the idea of eventually serving in wearable antennas in the context of smart-clothing applications. We particularly discuss the sensitivity of the design and fabrication of an inset-fed rectangular microstrip antenna (IRMA) in terms of its inset gap width when it is designed in the *S*-frequency band. The simulated and measured results showed a very small feasible range for the inset gap dimension with respect to the feed line width. Ultimately, an IRMA was successfully designed, fabricated, and tested with both SMA and U.FL connectors. The impedance bandwidth, in either case, was about 2%, the average value of directivity was 5.8 dB, and the realized efficiency was 2.67%, while the 3-dB beamwidths in the E-plane and the H-plane were 90${}^{{}^{\circ}}$ or wider.

## 1. Introduction

Microstrip antennas gained their popularity decades ago due to having a low profile, light weight, low manufacturing cost, and for showing the ability to be conformal and fit around curved bodies. Due to that, they are widely accepted for communications in various microwave frequency bands for terrestrial and satellite applications, mounted within mobile phones or wearable gadgets, some base station types, and on the surface of flying objects, despite the fact that they generally have low efficiency and gain due to their small size and thin substrate [1,2].

Implementing microstrip antennas for some wearable purposes, such as smart clothing [3], also calls for the antenna to be quite thin, flexible, and thermally robust, and to have a good moisture repellence.

To achieve the aforementioned traits, researchers have been trying various dielectric materials and compounds. In [4], a polydimethylsiloxane (PDMS) compound was tested as a flexible and water-safe antenna casing at 2.45 GHz, but it ended up being quite thick. In [5], a combination of acrylonitrile butadiene styrene (ABS) and NinjaFlex materials was tested at 2.3 GHz with a substrate height of 2.34 mm. In [6], a rubber substrate was tested at 2.45 GHz. In [7], 1.6 mm thick Arlon Diclade 1000 was tested as a flexible substrate for a 2.45-GHz antenna, and in [8], a polymeric substrate was tested as a flexible option for the antenna body. In the higher frequency bands, a 0.3 mm thin polymeric substrate was tested in [9] for an antenna at 28 GHz, while in [10], a combination of FR4 and Pyralux was utilized at 60 GHz, where FR4 was used in the middle as a substrate, while a Pyralux layer was used on the top and the bottom for the patch and the ground plane, altogether making the overall substrate thickness more than 1.75 mm.

Similarly to some of the above-cited works (e.g., [4,5,6]), we used a popular antenna model (described below) to examine another modern dielectric material (i.e., Pyralux^®^ by DuPont) with the idea of it eventually being used as a wearable antenna substrate. While a basic IRMA design has been described in textbooks [1,11] and some papers have discussed expressions that would calculate the inset length to match the particular input impedance of the antenna [12,13,14], a discussion on sizing the inset gap width is surprisingly hard, if impossible, to find within a systematic study. The case is that in most papers, IRMA has been built on a relatively thick substrate (such as FR4), and the inset gap width did not appear to be a sensitive variable to set up. However, when it comes to designing an IRMA on a Pyralux sheet as a substrate, we have not found enough verifiable (i.e., reproducible) reports that can be taken as a certain reference [15,16], especially using such a thin substrate on relatively low microwave frequencies such as the *S*-band. During this work, somewhat unexpected findings were encountered that we feel are relevant to share.

This paper contributes with the following: it offers a more comprehensive, sufficiently argued, and verifiable report on our experience with the design, optimization, fabrication, and measurement of and IRMA using Pyralux as a microstrip antenna substrate, both by full-wave simulations and measurements on fabricated models. Second, sizing of the inset gap width was found to be quite tricky at the combination of this frequency band and this ultra-thin a substrate, which makes it relevant to report on.

In our earlier works [17,18], we examined a circularly polarized wideband jeans-based antenna [17] and a circularly polarized multi-layer fully textile antenna made of common industrial fabrics and e-textiles [18]. Both of them were a few millimeters thick, semi-rigid (due to a multi-layer structure), and based on probe-fed excitation. At this point, we intended to examine the characteristics of an antenna when it is built on an ultra-thin, flexible, mechanically robust, and water-repellent substrate, such as Pyralux. It was also further sought that, in addition to using an ultra-thin substrate, the antenna should also attain the lowest possible profile, which led us to use an inset feed instead of a probe feed. Hence, we designed and fabricated an IRMA on a Pyralux-based substrate.

While Pyralux comes in multiple variants, Pyralux^®^ AP [19] was used in this work. Of its multiple options for the copper foil thickness and dielectric thickness, the AP9131R model was chosen, characterized by dielectric thickness h=75μm, copper foil thickness of 35μm, relative permittivity, $\epsilon_{r}=3.3$, and loss tangent, tanδ=0.0025. It is also an appealing option for in-house antenna fabrication because it enables processing of Pyralux sheets by conventional PCB fabrication techniques.

The rest of the paper is structured as follows: In Section 2, IRMA design expressions are recalled, including a short discussion about the design for a thicker substrate vs. a thinner substrate. In Section 3, we discuss early IRMA designs and our encountering of an unexpected difficulty with IRMA impedance matching, which required further investigation. Section 4 presents a successful IRMA design, together with its impedance bandwidth and radiation pattern. In Section 5, we discuss the encountered subtlety of sizing the inset gap width when an IRMA is designed in the *S*-frequency band on such a thin substrate. Lastly, Section 6 summarizes the work with the lessons learned and possible future work.

## 2. Design Considerations for a Rectangular Microstrip Antenna

### 2.1. Design Expressions

A layout of an inset-fed rectangular microstrip antenna (IRMA) is illustrated in Figure 1. While the design equations are widely available in the literature (with possible variations in some expressions), for completeness, we list those that were used in this work. The design frequency of 2450 MHz was chosen for a couple of reasons. One stems from the fact that our previous two textile antennas were designed for 2450 MHz, and that enables us a more direct comparison of the characteristics between the antenna models. Another reason lies in the fact that this frequency is in the ISM (Industrial, Scientific, and Medical) band, for which a number of commercial components are available and enable easier prototyping.

The patch width ($W_{p}$) and length ($L_{p}$) are calculated by [1]:(1)$$W_{p}=\frac{c}{2f_{r}\sqrt{\frac{\epsilon_{r}+1}{2}}}$$
(2)$$L_{p}=L_{eff}-2\Delta L$$
with
(3)$$L_{eff}=\frac{c}{2f_{r}\sqrt{\epsilon_{eff}}}$$
(4)$$\epsilon_{eff}=\frac{\epsilon_{r}+1}{2}+\frac{\epsilon_{r}-1}{2}{\left[1+12\frac{h}{W_{p}}\right]}^{-\frac{1}{2}}$$
and
(5)$$\Delta L=0.412h\frac{\left(\epsilon_{eff}+0.3\right)\left(\frac{W_{p}}{h}+0.264\right)}{\left(\epsilon_{eff}-0.258\right)\left(\frac{W_{p}}{h}+0.8\right)}$$

In (Equation 3)–(Equation 5), $L_{eff}$, $\epsilon_{eff}$, and ΔL are the effective patch length, the effective permittivity, and the length extension due to the fringing fields, respectively; other variables include the desired antenna resonant frequency ($f_{r}$), substrate height (*h*), substrate permittivity ($\epsilon_{r}$), and the velocity of light (*c*) [1].

The feed line width, $w_{f}$, is calculated by [11]:(6)$$B=\frac{377\pi }{2\;Z_{f}\sqrt{\epsilon_{r}}}$$
(7)$$w_{f}=\frac{2h}{\pi }\left[B-1-\ln\left(2B-1\right)+\frac{\epsilon_{r}-1}{2\;\epsilon_{r}}\left(\ln\left(B-1\right)+0.39-\frac{0.61}{\epsilon_{r}}\right)\right]$$
where $Z_{f}$ is the feed line characteristic impedance. Its total length, $L_{1}=L_{f}+y_{0}$, should be irrelevant as long as $Z_{f}$ equals the antenna input impedance, $Z_{A}$, but in later iterations, we kept it to be λ/2 long to make it more certain the $Z_{A}$ value will be mirrored to the front end of the feed line, which then enables more accurate transformation of the impedance to the 50-Ω connector impedance by the λ/4-transformer.

The role of the inset depth, $y_{0}$, is to connect the feed line to the desired antenna input impedance $R_{in}\left(y_{0}\right)$, knowing that $R_{in}$ decays from the patch edge towards the patch center. The calculation of $y_{0}$ has been covered by several sources [1,12,13,14], and the expression that was used here was [1]:(8)$$y_{0}=\frac{L_{p}}{\pi }cos^{-1}\left(\sqrt{\frac{Z_{f}}{R_{in}}}\right)$$
where $R_{in}$ is the antenna impedance at the patch edge, while the antenna impedance $Z_{A}=R_{in}\left(y_{0}\right)$ at the distance $y_{0}$ inside the patch is matched to the line impedance, i.e., $Z_{A}=Z_{f}$.

In contrast, the calculation of the inset gap value (*g*) has not been treated in the literature nearly as much (e.g., we found it in [20], but it did not show us promising values in this case). Due to that, *g* is typically chosen with the help of full-wave simulations, and in most cases it should be $g<w_{f}$; yet the question is what the ratio of $w_{f}/g$ should be for a successful design. A proper sizing of *g* turned out to be quite a delicate detail in the overall success of this antenna design and fabrication.

### 2.2. Difference with Respect to the Antenna Design on a Thicker Substrate

When an IRMA is designed on a thicker substrate, such as a widely used FR4 laminate, it results in a conveniently wide $w_{f}$ that can directly match the $Z_{A}$ at a 50-Ω point on the antenna patch. In contrast, the computation of the $w_{f}$ for its use with an ultra-thin substrate results in a $w_{f}$ value that is too narrow for a practical circuit and direct matching to a 50-Ω point on the patch. Consequently, when the $w_{f}$ value is set to be conveniently wide for use on a sheet of Pyralux, it results in a very low value of the feed line impedance and necessitates connecting it to the point on the patch where $Z_{A}=Z_{f}<<50\;\Omega$. That outcome then also necessitates the use of an impedance transformer, which matches the $Z_{f}$ to a standard 50-Ω connector. For that purpose, a quarter-wavelength (“λ/4”) transformer was chosen and designed as follows.

The necessary characteristic impedance value of the λ/4-transformer was obtained by the standard form [11]:(9)$$Z_{4}=\sqrt{Z_{1}\;Z_{0}}$$
where $Z_{1}$ is the impedance of the feed line at the intersection with the λ/4-transformer, and $Z_{0}=50\;\Omega$ is the impedance of the standard connector being connected to the front end of the λ/4-transformer. Having evaluated the $Z_{4}$, the necessary width of the λ/4-transformer, $w_{4}$, is then calculated by (Equation 6) and (Equation 7) when the $Z_{f}$ value is replaced in (Equation 6) by the value of $Z_{4}$ from (Equation 9). The necessary length of the λ/4-transformer, $L_{4}$, is given by
(10)$$L_{4}=\frac{\lambda_{0}}{4\;\sqrt{\epsilon_{4}}}$$
where $\lambda_{0}$ is the free-space wavelength, and $\epsilon_{4}$ is calculated by (Equation 4) when $W_{p}$ is replaced by $w_{4}$.

Next, we discuss how this theoretically clear concept did not turn out to be so straightforward to immediately achieve successful impedance and radiation characteristics of the fabricated antenna on this ultra-thin substrate.

## 3. The Pretested IRMA Models

The first IRMA design was based on the idea of having a reasonably wide feed line, $w_{f}=4.97\;\mathrm{mm}$, and the inset gap width, g=0.99mm, is conveniently sized for an in-house PCB-processing procedure. This setting makes $Z_{f}=3\;\Omega$, which is to be matched with the equal antenna input impedance. Using (Equation 8), $y_{0}=15.41\;\mathrm{mm}$. With $Z_{f}=Z_{A}$, the input impedance of the feed line, $Z_{in}$, is supposed to be equal to $Z_{A}$, while the feed line length, $L_{1}=L_{f}+y_{0}$ (according to Figure 1), should not matter and was arbitrarily set to be $L_{1}=30.41\;\mathrm{mm}$ (i.e., slightly shorter than λ/2), which gives $L_{f}=15\;\mathrm{mm}$. A λ/4-transformer (i.e., a pair of values $\left(L_{4},w_{4}\right)$) was then designed to match this low impedance to the 50-Ω impedance of the connector. However, the reflection coefficient of the fabricated circuit was unsatisfactory in spite of the reasonable theoretical basis and the promising results of the full-wave simulations [21].

So far, we kept $w_{f}/g=5$. It was considered sufficient, and earlier publications contained no clear recommendation on what the optimal ratio of $w_{f}/g$ should be. For example, in [22,23], $w_{f}/g<1$, but the designs were based on FR4. In contrast, in [24], $w_{f}/g=10$, while the circuit was also built on FR4. Of the few references that used Pyralux, in [15], $w_{f}/g=1.93$, while in [16], $w_{f}/g=0.73$. At this point, the optimal ratio of $w_{f}/g$ and the impact of *g* was not clear and had to be further investigated.

In the antenna model that was tried next, the feed line length was set to be exactly λ/2 to make it more certain that $Z_{A}$ would be mirrored to the front end of the feed line. However, the impedance matching was again unsatisfactory and very close to the previous result.

The next idea, to use the microstrip line both as the feed line and the λ/4-transformer, was not feasible because *g* would have to be prohibitively narrow and thus impossible to fabricate.

## 4. A Feasible Pyralux-Based IRMA

Up to this point, the $w_{f}/g$ ratio was kept equal to 5 and was considered to be a good enough ratio. However, based on our pre-tests and the discussion in [20], it was concluded that a larger $w_{f}/g$ ratio should be tried. To have a practically feasible *g* and the desired new ratio be equal to $w_{f}/g=10$, the feed line had to be wider and thus matched to an even lower antenna impedance, which was then chosen to be 1.5 Ω. The feed line length was again kept to be λ/2 long. The final design parameters of the IRMA are shown in Table 1.

### 4.1. The Impedance Bandwidth

Two versions of the Pyralux-based IRMA were fabricated—one with an SMA connector and another with a U.FL connector—for a comparison of their impedance-matching performance. The fabricated antennas are shown in Figure 2a, and their respective S${}_{11}$ parameters are in Figure 2b. It is evident that both connectors provide comparable performance. The measured resonant frequency of the SMA-based IRMA was 2397 MHz with a bandwidth of 49.8 MHz and fractional bandwidth (FBW) of 2.08%. The measured resonant frequency of the U.FL-based IRMA was 2381 MHz with a bandwidth of 47.9 MHz and FBW of 2.01%. The latter case is a preferred choice here because of its low profile. The U.FL-based IRMA resonant frequency was 2.82% lower than the design frequency of 2450 MHz, but such a discrepancy is quite common between the results achieved by analytical expressions, which help one make an initial design of the antenna, and the resonant frequency of the manufactured antenna. The simulated resonant frequency is also shown in Figure 2b and is located at 2.33 GHz, which makes just 2.1% difference between the simulated and the measured result for the U.FL-based IRMA. This discrepancy between the simulated and measured $S_{11}$ curves is practically irrelevant because what ultimately matters is the performance of the fabricated circuit, while the simulation is just an auxiliary tool to reduce the guess work. For example, when the circuit was simulated by another commercial full-wave solver [25], the simulated resonance appeared at 2.44 GHz, which is on the other side of the measured curves (two simulation tools—two different simulation results).

#### An IRMA with a Corrected Resonant Frequency

In the next iteration, we designed an IRMA for a resonant frequency that was 2.85% higher (i.e., designed for $f_{r}=2.52\;\mathrm{GHz}$) based on the initial result, which should compensate for the slightly lower value of the measured resonant frequency that was obtained in the preceding IRMA model. That compensation consequently resulted in the measured resonant frequency at 2433 MHz, which is very close to the original design frequency at 2450 MHz. Figure 3 shows that the new version of the IRMA exhibits an even better impedance match with −34 dB deep resonance at 2433 MHz. Due to that, the impedance bandwidth was slightly narrower than in the initial IRMA, exhibiting 43.2 MHz of absolute bandwidth or 1.77% of the fractional bandwidth.

The design parameters that were applied for this solution are listed in Table 2.

### 4.2. The Radiation Pattern

The radiation pattern was measured for the antenna with an SMA connector and a U.FL connector whose impedance characteristics are shown in Figure 2b. The SMA-based IRMA pattern was measured at 2397 MHz and showed directivity of D=4.8dB and realized efficiency of e=3.6%. The 3-dB beamwidths were 140${}^{{}^{\circ}}$ in the E-plane (i.e., YZ cut-plane) and 85${}^{{}^{\circ}}$ in the H-plane (i.e., ZX cut-plane). The U.FL-based IRMA was measured at 2381 MHz and had directivity of D=5.6dB but a lower efficiency of e=1.7%, with the 3-dB beamwidth of 90${}^{{}^{\circ}}$ and 95${}^{{}^{\circ}}$ in the E-plane and H-plane, respectively. It is noticeable that even though the frequency characteristics of the two antennas are almost equal (recall Figure 2b and the respective discussion in Section 4.1), there is a somewhat larger difference in their respective efficiencies of 3.6% and 1.7% and in the 3-dB beamwidth in the E-plane, which is 140${}^{{}^{\circ}}$ in the case of the SMA-based IRMA, while it is 90${}^{{}^{\circ}}$ in the case of the U.FL-based IRMA. In Figure 4, the measured radiation patterns are compared with the simulated radiation patterns and show good agreement.

In Figure 5, the measured radiation pattern of the U.FL-based antenna with the corrected resonant frequency is shown. It had D=7.1dB and e=2.7%, which are slightly higher values than in the case of the initial U.FL-based antenna due to better impedance matching. The radiation pattern of the frequency-corrected version of the U.FL-based antenna has identical 3-dB beamwidths as the initial U.FL-based IRMA, being 90${}^{{}^{\circ}}$ wide in the E-plane and 95${}^{{}^{\circ}}$ wide in the H-plane.

Even though this work does not propose a new antenna design but examines the effectiveness of a Pyralux sheet as a prospective antenna substrate, a comparison with some other references is presented in Table 3. We need to note that there were very few Pyralux-based antennas that we were able to find in the open literature and not many more references using other flexible substrates (except for various textile antennas, but they are not the focus here). Moreover, most of these papers do not specify all the antenna parameters that we declared for our antenna models. Due to this, we had to insert the “n/a” (i.e., “not available”) mark in some columns of Table 3. The columns of Table 3 compare: the center frequency ($f_{r}$), the substrate model and the height, the directivity or the gain (because different papers declared different quantities), the total efficiency or the radiation (“rad”) efficiency, the impedance bandwidth, and the beamwidths in the E-plane and the H-plane. We also note that although some papers did present the radiation pattern, they did not quantitatively specify the respective beamwidths, and we did not want to specify the numbers that were not typed in the original papers, but we put the “n/a” mark for such cases, too. Having said that, we see that our directivity is comparable to other references. Our bandwidth (BW) is similar to the declared BW of another Pyralux-based antenna [16] and the antenna with a polymeric substrate in [8] and is slightly smaller than that in [5,6,9], but we suspect that part of the lower value is also due to the use of a λ/4-transformer in our design. As can be seen in Table 3, most of the other references did not quantitatively specify the beamwidths obtained by their antenna models, while our beamwidths were equal or wider than 90${}^{{}^{\circ}}$ in both major cut-planes.

## 5. Discussion

The small impedance bandwidth and the low radiation efficiency can be attributed to the ultra-thin substrate and the very low dielectric losses of Pyralux™AP9131R. While such characteristics of the substrate are generally not a preferred choice when it comes to antenna design in general [1], testing the antenna on this type of a substrate is justified in the context of a smart-clothing application and the desired seamless integration of an antenna into a garment.

Unlike antennas manufactured with a standard substrate, such as FR4, which show a wider feasible range of the inset gap width values, this thin a substrate exhibited a very small range of acceptable values of $w_{f}/g$ to make good impedance matching, at least in this frequency range. In the first designs, we hoped that $w_{f}/g=5$ was an adequate ratio and searched for some other reasons that would have explained the unsatisfactory impedance matching. The CAD models also required mesh refinement until more confident results were achieved. The reason stems from the fact that Pyralux AP9131R is about 20 times thinner than typical substrates such as FR4 or RT/duroid. Ultimately, when the $w_{f}/g$ ratio was sufficiently increased, a satisfactory reflection coefficient was achieved, as shown in Figure 2b.

Finding a nearly optimal *g* was not a straightforward task because: (a) there is no widely adopted formula that helps calculate an optimal *g*; and (b) the ratio of $w_{f}/g$ is quite different in different papers (assuming the results presented there were entirely trustworthy). The results of a sweep analysis, which we undertook using a full-wave simulator [21], are shown in Figure 6 and indicate that there is some optimal ratio of $w_{f}/g$, while higher and lower values of $w_{f}/g$ deteriorate the impedance matching.

The curves in Figure 6 are the results obtained for the values of *g* and the ratios of $w_{f}/g$ that are listed in Table 4.

It is significant to notice that the $w_{f}/g$ ratio substantially affects impedance matching, even for a small change of *g* value. One can clearly notice it by observing the $S_{11}$ results in Figure 6 for g=0.84mm vs. g=0.67mm vs. g=0.5mm. However, $w_{f}/g$ also substantially affects the resonant frequency value! That finding ultimately explains why poor impedance matching results were achieved in the first designs where $w_{f}/g=5$ was used (the green curve in Figure 6).

As for the fairly narrow impedance bandwidth that was achieved, we believe that it cannot be attributed merely to the properties of Pyralux (for being ultra-thin and low-loss), but is also due to the nature of the λ/4-transformer, which exhibits quite a narrow impedance bandwidth when matching two impedances whose ratio exceeds 10.

## 6. Conclusions and Future Work

In this work, we tested the performance of a Pyralux laminate when serving as a substrate for a microstrip antenna that was designed to operate in the *S*-frequency band around 2.45 GHz. An IRMA, as a popular antenna model, was chosen to support this study.

Narrow bandwidth and low efficiency were expected due to the Pyralux characteristics that were described in the paper, while such a narrow margin for a feasible and successful design, subject to having the ratio of $w_{f}/g=10$ and dealing with very low antenna input impedance, was not expected. At last, we still achieved typical radiation patterns of an IRMA, with an average value of directivity equal to D=5.8dB and beamwidths wider than 90${}^{{}^{\circ}}$ in the E-plane and the H-plane. Impedance matching was adequate in terms of the reflection coefficient, with $|S_{11}|\leq -16\;\mathrm{dB}$, while realized antenna efficiency and impedance bandwidth achieved lower values than desired. In future iterations, it is worth trying different impedance-matching circuits and perhaps different excitation techniques with the aim of widening the bandwidth.

An antenna built on this type of a laminate is still an interesting option to be further investigated in the context of wearable antennas and their seamless integration into a garment.

## Figures and Tables

**Figure 1 sensors-23-00213-f001:**
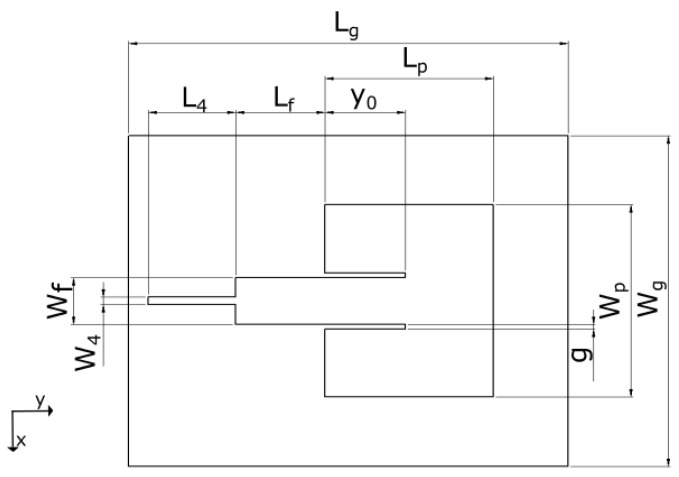
General layout of an inset-fed rectangular microstrip antenna.

**Figure 2 sensors-23-00213-f002:**
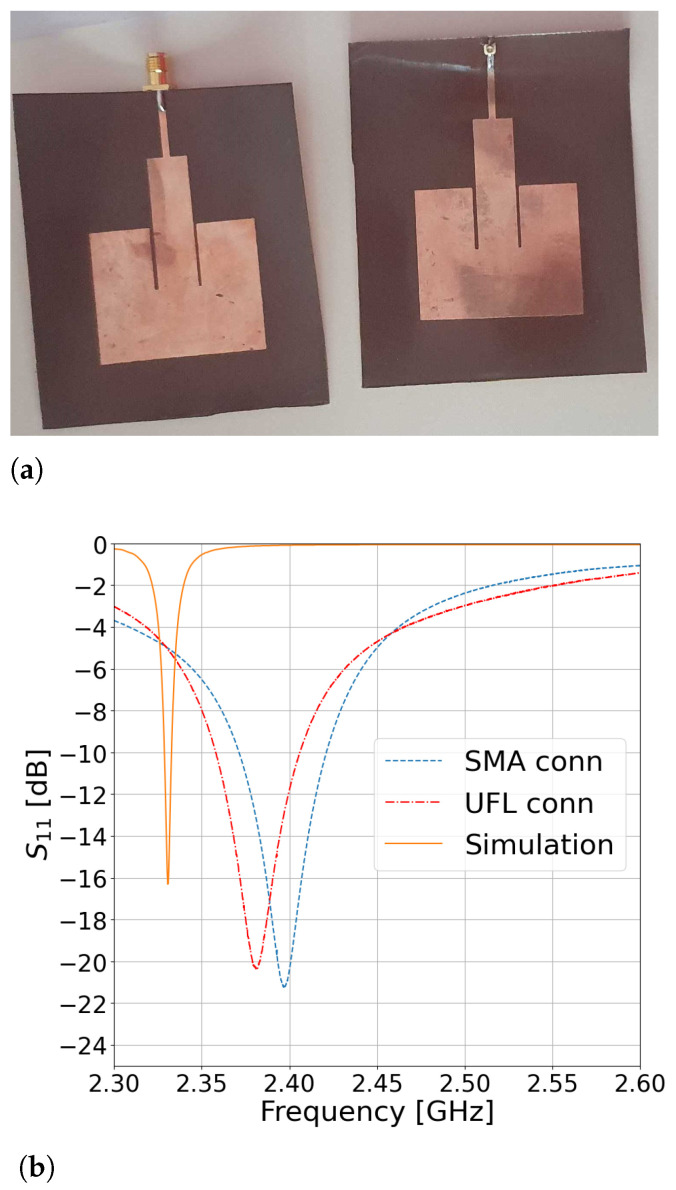
The feasible IRMA models: (**a**) fabricated with an SMA and a U.FL connector; (**b**) the S${}_{11}$ parameters.

**Figure 3 sensors-23-00213-f003:**
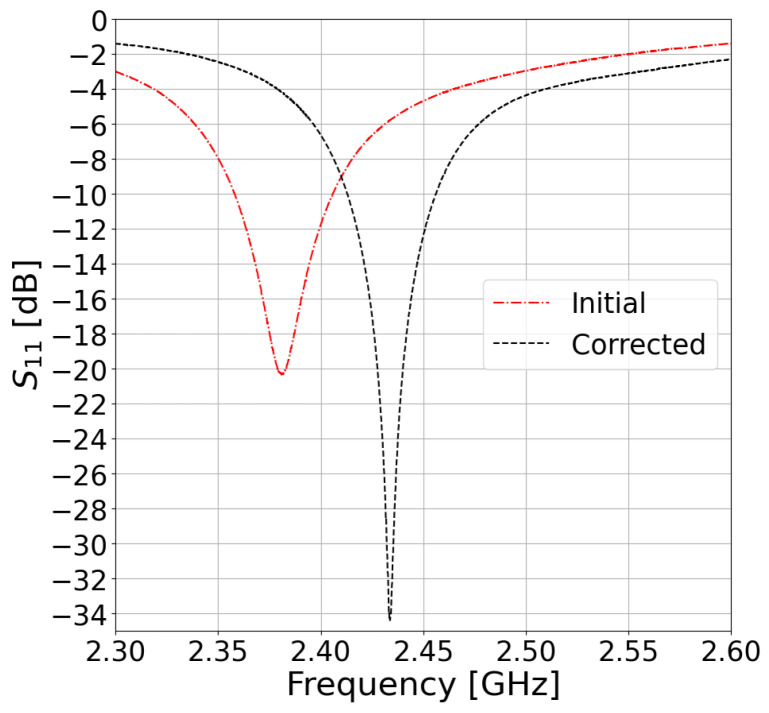
The measured resonance of the U.FL−based IRMA with the resonant frequency-compensated design.

**Figure 4 sensors-23-00213-f004:**
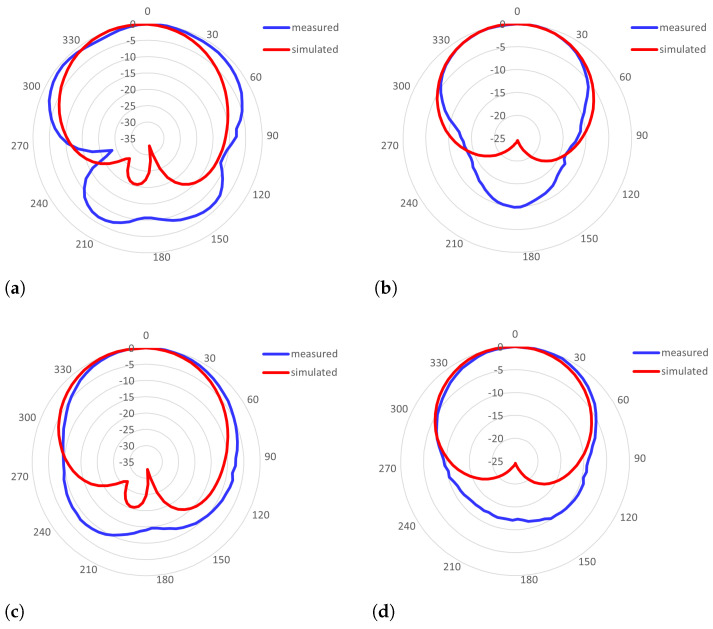
The simulated vs. measured radiation patterns of the Pyralux-based IRMAs with: (**a**) an SMA connector: E−plane; (**b**) an SMA connector: H−plane; (**c**) a U.FL connector: E−plane; and (**d**) a U.FL connector: H−plane.

**Figure 5 sensors-23-00213-f005:**
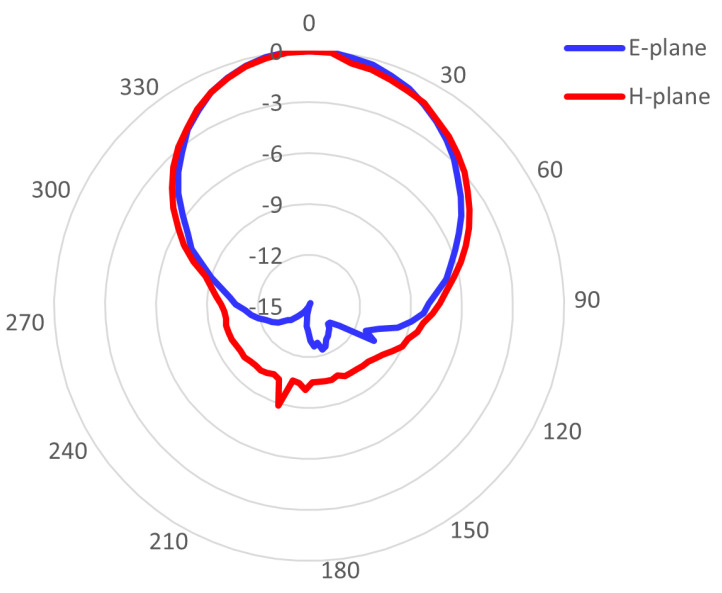
The measured radiation pattern of the resonant−frequency−corrected U.FL-based antenna.

**Figure 6 sensors-23-00213-f006:**
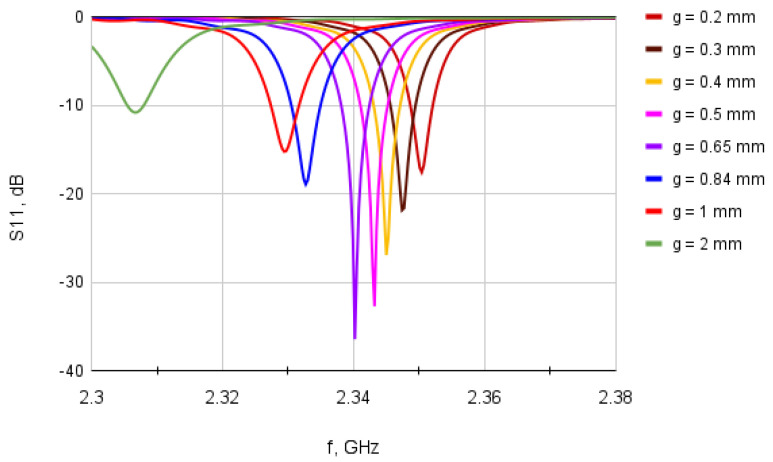
A sweep analysis showing dependence of S${}_{11}$ versus *g*.

**Table 1 sensors-23-00213-t001:** The final design parameters of the fabricated IRMA.

Parameter	Value (mm)
$L_{p}$	33.69
$W_{p}$	41.75
$L_{g}$	88.05
$W_{g}$	71.75
$y_{o}$	16.06
*g*	1.01
$w_{f}$	10.13
$L_{f}$	17.89
$w_{4}$	1.61
$L_{4}$	17.47

**Table 2 sensors-23-00213-t002:** The design parameters of the frequency-corrected IRMA.

Parameter	Value (mm)	Parameter	Value (mm)
$L_{p}$	32.76	*g*	1.01
$W_{p}$	40.59	$w_{f}$	10.13
$L_{g}$	86.13	$L_{f}$	17.39
$W_{g}$	70.59	$w_{4}$	1.61
$y_{o}$	15.62	$L_{4}$	16.98

**Table 3 sensors-23-00213-t003:** Characteristics of some other IRMAs.

Ref.	$f_{r}$ (GHz)	Substrate					HPBW (${}^{{}^{\circ}}$)	
		Model	Height (mm)	D or G (dBi)	Effic (%)	BW (%)	E-Plane	H-Plane
[5]	2.3	flexible ABS, Ninja-Flex	2.34	n/a	n/a	5.21	n/a	n/a
[6]	2.45	rubber (flexible)		3.42 (G)	60 (rad)	4.1	n/a	n/a
[8]	2	polymeric, flexible		2.55 (G)	84 (rad)	2.6	n/a	n/a
[9]	28	polycar-bonate	0.3	5.68 (G)	69	4.2	87	n/a
[15]	2.13	Pyralux	0.0508	−1.8 (D)	n/a		n/a	n/a
[16]	3.29	Pyralux FR9111	0.025	1.4 (G)	n/a	2.4	n/a	n/a
this	2.38	Pyralux AP9131	0.075	5.8 (D)	2.67	2	≥90	≥90

Column labels—*f_r_*: center frequency; D or G: directivity (D) or gain (G), depending on what was declared; effic: total efficiency or radiation efficiency (rad); BW: impedance bandwidth; and HPBW: −3-dB beamwidth.

**Table 4 sensors-23-00213-t004:** The feed line width ($w_{f}$) versus the inset gap width (*g*).

***g* (mm)**	2	1	0.84	0.65	0.5	0.4	0.3	0.2
$w_{f}/g$	5.07	10.13	12.06	15.58	20.26	25.33	33.77	50.65

*with**w_f_* = 10.13 mm *in all cases*.

## Data Availability

Not applicable.

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
