# Peer review of "A Subtlety of Sizing the Inset Gap Width of a Microstrip Antenna When Built on an Ultra-Thin Substrate in the S-Band"

_sensors, 2022, doi:10.3390/s23010213_

Round 1

Reviewer 1 Report

In this research paper, inset fed microstrip patch antenna is investigated. This paper needs a major revision as follows;

1. The originality of the paper is missing, please explain the novelty of the study.

2. Introduction section does not give a comprehensive literature review, please cite more studies from recently published research articles.

3. Please explain why this operating frequency has been chosen.

4. Figure 6 is not in a good format, please give more suitable figure format.

5. The simulated radiation patterns are missing, it is best to give both simulated and measured results in the same figure.

6. If possible, please give a literature comparison in a tabular form.

7. The main idea is not clearly explained, please explain the theory by more detailed mathematical formulas.

Author Response

Dear Reviewer,

Please find the complete Response to your review attached as PDF file. 

Thank you for your time!

Miroslav Joler

Reviewer 2 Report

An inset-fed rectangular microstrip antenna (IRMA) was designed, fabricated and tested with SMA and U.FL connectors on an ultra-thin substrate. The sensitivity of the design and fabrication of the antenna in terms of the inset gap was discussed. However, there are some questions and comments regarding to the manuscript as follows:

1- The novelty of the manuscript is poor. A lot of research has been done relating to the patch antennas, and inset feeding.

2- The main goal of the authors is discussion about the sensitivity of the design and fabrication of the antenna in terms of the inset-gap. However, no method or design rules are provided.

3- Usually, radiation pattern and 3-dB beam-widths are presented in E and H-planes. The phrase "in each cut-plane 90◦" in the abstract is not correct.

4- Literature review and Introduction is not complete.

5- The feeding connector of the antenna can affect the input VSWR, but has no significant effect on the radiation pattern. So, what is the reason for the significant difference between the Fig. 4a, and Fig. 4b?

6- For an in-house PCB-processing procedure, "g" was selected to be 1.01 mm. This is a large gap-width, implies a large value for "Wf", which affect the antenna radiation pattern. In addition, the antenna may have another resonance frequency.

7- The paper should be revised for English writing and grammar errors.

Author Response

(The authors gave the same response as above.)

Reviewer 3 Report

The Manuscript describe about the Microstrip Antenna. Below are my comments:

Shorten the title. It seems to be not nice. Too long title.

Abstract need to revised with grammatical correction. Why is the directivity so less?

Manuscript need to be heavily revised with author details. Current address is missing.

I cannot understand line 32 to 35. Revise it. Introduction is too few. Perform more research study.

Too poor writing from line 59 to 69. Revise it.

Line 82 to 94?? Its ambiguous.

Revise line 103 to 116. Manuscript need a lot of grammatical sentence correction.

Why there is large shift in frequency against the simulated results in figure 2(b)?

Plot the figure from Matlab. The label in Figure 3 and 2 need to be revised. Especially Y-axis values.

Plot the 2D radiation plots. Polar plot is not convincing. Which one is H- and E- planes??

Revise Figure 6 with Matlab plot.

Provide the comparison table against the other proposed methods.

In overall, the manuscript is like difficult to understand in terms of grammatical correction.

Need to have English correction. Then revise the plots as suggested.

Author Response

(The authors gave the same response as above.)

Round 2

Reviewer 1 Report

This paper is well revised and can be acceptable for Sensors in this revised form.

Author Response

Dear Reviewer,

Thanks!

Reviewer 2 Report

The manuscript have been revised based on the comments.

New comments:

1- In the NTRODUCTION, insert the complete phrase for "IRMA".

2- In section 4, there is following sentence: "However, based on our pretests and the discussion in [20], it was concluded that a larger wf /g ratio should be tried." It seems that Ref. [20] is not for you. Please check the Ref. numbers.

Author Response

Dear Reviewer,

Thanks!

Reviewer 3 Report

The way author has revised manuscript is not suitable.

Literally revised your abstract. I feel to confused to read your abstract. Lots of grammatical correction is necessary in sentence structures.

Rephrase the line 49 and 50. Somewhat unexpected finding!!!!

Line 51 to 57 can be kept in conclusion section.

It was an intention to examine the an antenna characteristics??? Is this correct sentence.

Which then led us to using a ??? Is this correct sentence.

Of its multiple?? Is this correct way to start the sentence?

Line 125-126, 135-137, 153-155 I am really in trouble to understand your flow of language and sentence structure. Better to revise it through some professional language editor.

I don’t think that just direct plot of figure from CST or HFSS is good.

The polar plot label that is the numbers is not clear.

The Table is not organized properly. The D and G are kept in same column. this can be replaced by proposed work. Check the literature papers how to frame the comparison table.

Provide me the clear manuscript. Conclusion should be short and precise. You have just exaggerated much information. 

Author Response

Dear Reviewer,

Thanks!
